

# Application of artificial intelligence technology in the economic development of urban intelligent transportation system

Ziming Zhao[1] and Jinyu Chen[2]

[1] School of Urban Economics and Public Administration, Capital University of Economics and Business, Beijing, China
[2] College of Business Administration, Capital University of Economics and Business, Beijing, China

## ABSTRACT

With the rapid development of the social economy and the gradual improvement of residents' living standards, the increasing number of urban cars has exacerbated urban traffic congestion. This article analyzed the application of artificial intelligence (AI) technology in five aspects of urban intelligent transportation systems. Artificial intelligence technology was used in traffic data collection and processing to provide accurate data support for traffic decision-making. A traffic flow prediction model was established for traffic flow prediction and optimized scheduling algorithms were used to dispatch vehicles on congested urban roads intelligently. Artificial intelligence algorithms can be used to optimize urban traffic signal control systems in intelligent traffic signal control; artificial intelligence technology can be applied to develop intelligent driving systems in the fields of intelligent driving and traffic safety; in terms of data analysis and decision support, it can use AI technology to analyze a large number of traffic data to provide decision support for urban traffic managers, and analyze the impact of the application of AI technology in urban intelligent transportation system on urban economic growth. This article evaluated the economic benefits of artificial intelligence technology in urban intelligent transportation systems. The evaluation results show that the total economic cost of the urban intelligent transportation system after the application of AI technology was 2,961 yuan less than before the application of AI technology, significantly reducing the investment cost of roads. This article analyzes the application of artificial intelligence technology in the economic development of intelligent urban transportation systems, which can meet the needs of healthy urban development and ensure road traffic safety.

# INTRODUCTION

With the rapid development of electronic communication technology and the acceleration of urbanization construction, artificial intelligence (AI) technology based on sensors and control systems is widely used in urban infrastructure. The application of artificial intelligence technology has facilitated urban transportation and reduced energy

Corresponding author
Ziming Zhao,
sjmzhaoziming@163.com

consumption. An urban intelligent transportation system has become an essential part of modern urban development. This article analyzes the economic development application of AI technology in urban intelligent transportation systems. It discusses the impact of AI technology on urban transportation efficiency, financial benefits and sustainable development.

Because the current scientific and technological development is imperfect, the development of urban intelligent transportation systems (ITS) faces many difficulties. Experts have conducted an in-depth analysis of this. *Veres & Moussa (2019)* believed that the challenges faced by modern ITS are many spatial and temporal characteristics on different scales under different conditions brought about by external sources such as social events, holidays and weather. Based on this, they modeled the interaction of factors, designed generalized representations, and then used them to solve specific problems (*Veres & Moussa, 2019*). *Yu et al. (2020)* believed that with the increasing complexity of the intelligent transportation system, autonomous vehicles show low intention recognition rate and poor real-time performance when predicting the driving direction, which seriously affects the safety and comfort of the hybrid transportation system. *Ferdowsi, Challita & Saad (2019)* believed the intelligent transportation system would become essential to the future smart city. However, realizing the real potential of the intelligent transportation system requires ultra-low delay and reliable data analysis solutions. These solutions can combine heterogeneous data from the ITS network and its environment in real time (*Ferdowsi, Challita & Saad, 2019*). *van der Heijden et al. (2018)* believed that a collaborative intelligent transportation system is a promising technology that can improve driving safety and efficiency. Vehicles can create a highly dynamic and heterogeneous management self-organizing network through wireless communication with other vehicles and infrastructure (*van der Heijden et al., 2018*). The analysis of the current development status of urban intelligent transportation systems shows that the development of modern intelligent transportation systems is greatly hindered under the influence of external conditions and technical conditions.

*Choudhury et al. (2025)* examined the revolutionary power of machine learning (ML) in transportation coverage and showed how it enables improved route planning, better tracking, and accident prevention capabilities. They outlined how Internet of Things (IoT)-driven data collection continues to rise because it makes smarter transportation systems possible. The review acknowledged ML's promising applications in the industry but recognized privacy issues that will obstruct progress in future developments. Specifically for transportation applications, ML was marked as essential for operational transformation, yet it encountered several development obstacles that need to be resolved in the future. *Panda et al. (2025)* studied cloud computing applications for urban transportation systems and evaluated how this technology solves transportation-related issues, including traffic congestion, infrastructure limitations, and environmental pollution. Cloud-based smart transportation solutions optimize urban mobility through IoT devices, data analytics, and real-time monitoring operations. They stressed how cloud platforms maintain high scalability and cost-effectiveness with flexible features for efficient data collection, storage, and analysis. These solutions' better decision-making capabilities

enable better traffic management, reduced congestion, and improved urban safety and sustainability. *Sarwatt et al. (2024)* conducted a detailed examination of the metaverse's potential impact on ITS by studying future technology possibilities such as virtual reality, digital twins, blockchain, and AI to manage current transportation problems, including security, privacy issues, and data management challenges. A recent survey demonstrated that the metaverse provides several groundbreaking solutions by integrating secure communication tools with virtual testing platforms and a centralized system for real-time data processing. Although ITS has yielded major development milestones, this review showed researchers still need to investigate the unrevealed possibilities of metaverse solutions in this area. They presented different real-world applications and economic and societal implications of metaverse use as parts of their research to show how it can transform transportation systems in the future.

With the development and maturity of artificial intelligence technology, it has been applied to various fields and is also widely used in transportation. Many experts have analyzed the specific applications of artificial intelligence technology in transportation. *Janowicz et al. (2020)* believed that the latest progress in artificial intelligence technology, the large-scale availability of high-quality data, and the progress in hardware and software for effectively processing these data are changing the field from computer vision and natural language processing to automatic driving. *Paschen, Pitt & Kietzmann (2020)* believed that an example of product-oriented innovation in artificial intelligence is the rise of autonomous vehicles in urban transportation. In contrast, an instance of process-oriented innovation is using artificial intelligence technology to innovate in all aspects of transportation (*Paschen, Pitt & Kietzmann, 2020*). *Wirtz, Weyerer & Geyer (2019)* believed that the rapid development of automation and artificial intelligence would significantly impact the transportation field. They studied the safety issues related to artificial intelligence in transportation and focused on applying artificial intelligence in traffic risk prevention strategies (*Wirtz, Weyerer & Geyer, 2019*). By analyzing the application of artificial intelligence technology in the field of transportation, it was found that artificial intelligence technology is mainly used in autonomous driving and traffic risk prevention. Still, its impact on urban economic development has not been thoroughly evaluated and analyzed.

With the development of the urban economy, many people have flooded into cities and towns, resulting in a gradual increase in vehicles, population, and buildings in cities, leading to increasingly serious urban traffic congestion. The research direction of many experts is on how to use modern technology to solve urban traffic congestion. Facing the new method in the interconnected environment, *Sumalee & Ho (2018)* analyzed the economic benefits by understanding how the current urban intelligent transportation system adapts to the interconnected environment. He introduced a new method of real-time flexible control and management of the transportation system to improve the economic benefits of the current urban intelligent transportation system (*Sumalee & Ho, 2018*). *Jan et al. (2019)* believed that traffic congestion directly impacts the social and economic activities of cities. Through big data technology analysis, they applied to efficient design and planning of intelligent transportation, intelligent control systems, smart cities,

smart communities and other aspects, which can improve the overall economic benefits of the city (*Jan et al., 2019*). Research on the economic benefits of intelligent urban transportation systems has found that the quality of traffic management dramatically impacts the development of the urban economy. It is necessary to build an urban intelligent transportation system through modern new technology to reduce traffic congestion.

Modern trends in artificial intelligence have a profound impact on the growth of urban ITS. *Veres & Moussa (2019)* focused on the application of deep learning for ITS with the potential to address spatial and temporal issues related to real-time traffic management. In the same way, *Yu et al. (2020)* discussed AI-based traffic safety proposals regarding the hybrid transportation system, with the intervention of deep learning for improving the safety features of both the autonomous and manual vehicles in 5G ITS. In particular, *Ferdowsi, Challita & Saad (2019)* underlined how mobile edge analytics is instrumental in supporting the ITS to perform real-time, accurate data analysis about the integration of heterogeneous data to improve transportation systems. On the application side, they envisioned that GeoAI methods can enhance the management of geographic knowledge to enhance navigation and infrastructure. They also divided the AI innovations into the product and process AI innovations; a sample highlighted by the authors includes self-driving vehicles and AI-based traffic control systems. Despite these advancements, there is still a shortfall of practical research on the economic perspectives of AI in ITS. Previous research mainly targets technical and operational enhancement aspects, while the economic consideration aspect is relatively highlighted. This article fills this gap by examining the economic feasibility of AI in the context of ITS for sustainable development of modern urban societies and subsequent economic growth (*Wirtz, Weyerer & Geyer, 2019*).

AI and traffic systems in future smart city plans will reportedly be assimilated into other systems, such as energy networks and water. For instance, traffic information can regulate energy supply by responding to peak traffic hours or synchronizing water consumption for car washing and other municipal services. Integrating these systems encourages an integrated solution to manage a city's functions by improving its efficiency, sustainability, and general toughening of all systems.

# APPLICATION OF ARTIFICIAL INTELLIGENCE IN TRANSPORTATION

## Data collection and processing

### Process of collecting data

This article uses artificial intelligence technology to collect, process, and analyze urban traffic data to provide accurate data support for traffic decision-making. Sensors and wireless communication technology are used to collect on-site traffic data. The specific information collection process is shown in Fig. 1.

Figure 1 demonstrates the traffic information collection process based on a randomly captured traffic map of an intersection in a certain city. The red instrument is a sensor that senses and converts the physical and chemical quantities in the surrounding environment

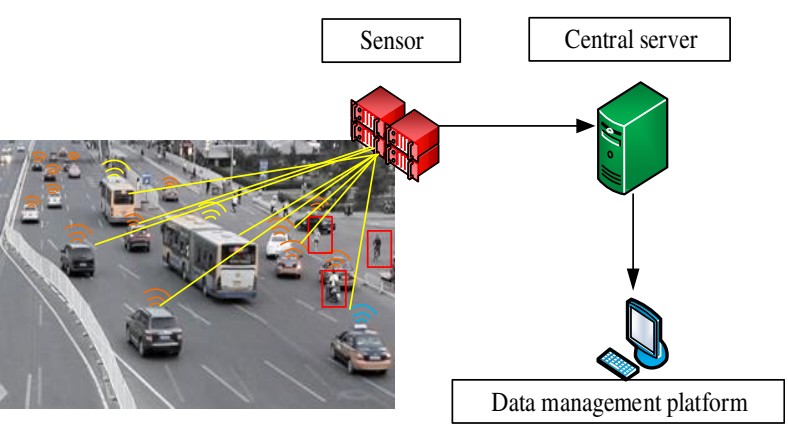

**Figure 1  Flow chart of traffic data collection.**

of the intersection. Various signals or information at intersections can be transformed into electrical or other signals that can be detected, recorded, processed, and utilized. This article mainly analyzes vehicles in motion, and in response to the difficulty of collecting data information during vehicle movement, sensors are used primarily for object motion and position detection. It can provide helpful information and signals by transforming various data of moving objects into intelligent decision-making in intelligent transportation systems. In addition, the green instrument is the central server, which plays a core role in the intelligent transportation system. A large number of vehicles pass through the intersection every day. Therefore, huge data would be generated. At this time, the central server is required to store and manage data in a centralized manner, calculate and process data, manage and control the network, share and coordinate resources, manage security and permissions, and backup and restore data to provide support for the stable operation and efficient management of the intelligent transportation system. Finally, the blue instrument is the data management platform, which can effectively help the intelligent transportation system management to use the data processed by the central server and has a massive role in improving the data collection technology of the intelligent transportation system in the later stage.

### Processing collected data information

Through the process of collecting data in Fig. 1, it is possible to collect various information on the vehicles driving on the road. Through data collection, it is possible to effectively grasp the traffic flow, vehicle, and road condition information of the road. Adjustments can be made in a timely manner to avoid traffic congestion, as shown in Fig. 2.

Figure 2A is the traffic flow information map of the intersection during the morning rush hour, where the horizontal axis represents the vehicle type. There are three types of car models, namely taxis, sedans, and buses. The vertical axis represents the quantity, which means the number of each type of car during the morning rush hour at an intersection. From Fig. 2A, it can be clearly seen that the number of vehicles is the highest, while the number of buses is the lowest. This indicates that residents who travel in the area in the morning tend to drive their cars. Figure 2B is the traffic flow information map of the

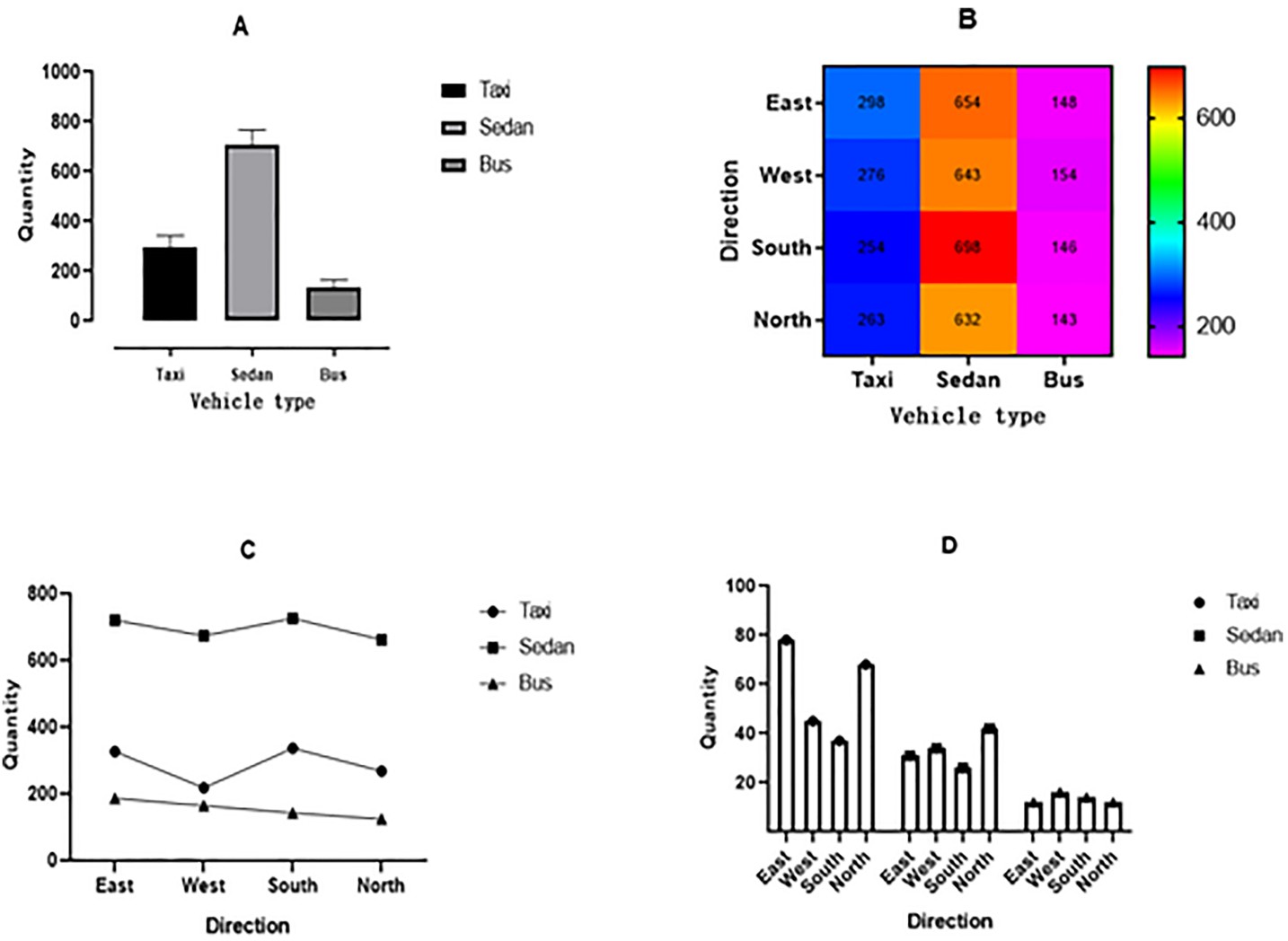

**Figure 2 Traffic flow information map of a certain intersection during morning, noon, evening, and night rush hours.**

noon peak at the intersection, where the horizontal axis represents the vehicle type, which is mainly divided into three types: taxi, sedan, and bus. The vertical axis represents the direction in which the vehicle enters the intersection. Four entrances and exits at the intersection correspond to four directions: east, south, west, and north. Figure 2B shows that the number of small cars is also the highest during the afternoon rush hour, while the number of large buses is the lowest, indicating that there are more models of small and medium-sized cars on urban roads.

Figure 2C is the traffic flow information map of the intersection during the evening rush hour, where the horizontal axis represents different directions. The vertical axis represents the number of vehicles in each direction. Figure 2C shows that the number of buses entering from the north entrance is less than that from the east entrance, while the number of taxis entering from the north entrance and west entrance is less than that from the east entrance and south entrance. Similar to taxis, cars also have fewer entrances from the north

and west entrances than from the east and south entrances. Figure 2D shows the traffic flow information map of the intersection at night. It can be clearly seen that the number of three models is significantly reduced compared to Figs. 2A–2C. This is because the evening is a time for residents to rest, and it is more evident from the parallel comparison of the three models that the number of buses is the least.

In conclusion, the intelligent transportation system can monitor and analyze road traffic in real-time through data collection and processing. The data collected by devices such as traffic sensors and cameras can be used for real-time monitoring of congestion conditions, traffic flow, vehicle speed, and other indicators, and based on this data, timely traffic scheduling and management decisions can be made. Data collection and processing can provide critical traffic information, enabling traffic managers to formulate targeted traffic strategies and measures to optimize the utilization of road resources, reduce congestion, and improve traffic efficiency. In addition, through analyzing and processing traffic data, the intelligent transportation system can detect abnormal conditions and send out accident and traffic safety warnings. In a word, the intelligent transportation system's data acquisition and processing technology can provide accurate and practical data support for traffic managers, drivers, and urban planners to improve traffic conditions and the efficiency and safety of the transportation system.

### Hardware and infrastructure requirements

Proper implementation of artificial intelligence traffic monitoring calls for strong hardware and a base infrastructure to support a large number of systems. Some of these components include high-definition and real-time cameras for capturing video data, IoT devices for capturing real-time data on traffic and climate, and edge devices for real-time data processing to minimize on delay. Protocols like 5G are used in connecting devices to enhance the transfer of data between gadgets and some mainframes at the shortest time possible. Moreover, cloud-based solutions give technological support for storing and processing big data to integrate into one platform. Other aspects include compatibility with existing systems, including traffic signals and vehicle detectors, in order to allow the integration. These parts make up the foundational elements of AI-centred traffic systems so that accurate monitoring, analysis, and decision-making can take place throughout cities.

## Traffic flow prediction and scheduling

This article uses artificial intelligence technology to establish a traffic flow prediction model, accurately predict urban traffic flow, and optimize scheduling algorithms to alleviate traffic congestion and efficiently utilize traffic resources.

### Traffic flow prediction model

Accurately predicting the traffic capacity of urban roads can effectively dispatch congested sections, alleviate traffic pressure on congested sections (*Shengdong, Zhengxian & Yixiang, 2019*), and save residents' fuel and time costs for travel. Therefore, this article uses artificial intelligence technology to establish a traffic flow prediction model using the support vector

machine (SVM) regression algorithm in machine learning algorithms to achieve prediction of urban traffic flow (*Saleem et al., 2022*).

First, define nonlinear mapping:

$$\partial(\cdot): \theta^m \to \theta^m j. \tag{1}$$

Next, input the training dataset:

$$\{(C_o, u_o)\}_{o=1}^{M}. \tag{2}$$

Finally, map to $\theta^m j$.

In the plane of the dataset, there is a non-linear relationship between the input and output data, denoted as g, and the support vector regression function is:

$$g(C) = E^Y \theta(C) + n. \tag{3}$$

Among them, g(C) is the predicted value, coefficient $\theta(\theta \in \theta^m j)$ and coefficient $n(n \in \theta)$ are variable values.

Support vector regression is used to minimize empirical risk minimization, as shown in the following formula:

$$T_{emp}(g) = \frac{1}{M} \sum_{o=1}^{M} \mu_\varepsilon\left(y_o \delta^E \gamma(c_o) + n\right). \tag{4}$$

$\mu_\varepsilon(y, \gamma(c_o))$ is an insensitive Loss function, and $\epsilon$ is defined as follows:

$$\mu_\varepsilon(y, g(c)) = \begin{cases} |g(c) - y| - \varepsilon & |g(c) - y| \geq \varepsilon \\ 0 & |g(c) - y| < \varepsilon. \end{cases} \tag{5}$$

$\mu_\varepsilon(y, g(c))$ searches for the optimal hyperplane in the space of high-dimensional features and divides the input training data into two sides of the hyperplane. The Hyperplane is two lines. These two lines can be denoted as $-\epsilon$ and $+\varepsilon$, respectively, to facilitate the formation of the maximum distance between subsets in the training data. Then, when training the data in the support vector machine regression model, the training error is minimized, as shown in the following formula:

$$\min T_\epsilon(\delta, \vartheta^*, \vartheta) = \frac{1}{2} \delta^Y \delta + V \sum_{o=1}^{M} (\vartheta_o^* + \vartheta_o). \tag{6}$$

The constraint conditions of the above formula are:

$$\begin{aligned} y_o - E^Y \theta(c_o) - n \leq \varepsilon + \vartheta_o^*, o = 1, 2, ..., M, \\ -y_o + E^Y \theta(c_o) + \varepsilon + \vartheta_o, o = 1, 2, ..., M, \\ \vartheta_o^* \geq 0, o = 1, 2, ..., M, \\ \vartheta_o, o = 1, 2, ..., M,. \end{aligned} \tag{7}$$

The insensitive Loss function minimizes the data training errors of $g(c)$ and y. The errors on both sides of the hyperplane are expressed with different parameters. The training errors above $+\varepsilon$ are expressed as $\vartheta_o^*$, and the training errors above $-\varepsilon$ are expressed as $\vartheta_o$.

By optimizing the parameter vector $\delta$ twice, the following formula can be obtained:

$$\delta = \sum_{0=1}^{M} (\alpha_0^* - \alpha_0)\theta(c_o) \tag{8}$$

$\alpha_0^*$, $\alpha_0$ are the Lagrange multiplier obtained through secondary optimization, and finally the regression function of the support vector machine regression model is obtained in the dual space, as shown in the following formula:

$$g(c) = \sum_{0=1}^{M} (\alpha_0^* - \alpha_0)L(c_o, c) + n. \tag{9}$$

The traffic flow prediction in this investigation uses the SVM regression algorithm. The decision trees and neural networks were not selected instead of SVM because of the latter's efficiency in small and medium sample size data sets for the non-linear relationships in data and generalization. On the one hand, decision trees have the benefits of being interpretable and having low computational complexity. Still, on the other hand, they lead to overfitting, especially in cases of high traffic complexity. On the other hand, ITS applications that involve neural networks need much larger datasets and computational horsepower, which does not fit ITS's real-time needs. An efficient mathematical tool in SVM regression transforms nonlinear data into a higher-dimensional space for minimal empirical risk for accurate predictions. Due to its application of the $\varepsilon$-insensitive loss function, minor changes in prediction can occur without much affecting the model. Furthermore, the optimization of SVM makes it possible to define a hyperplane with maximum margin, which is the method's main advantage when it is used to predict traffic flows in cases where nonlinear dependencies are observed.

In summary, this article uses the support vector machine regression algorithm to map the trained nonlinear data into an infinite dimensional space, which can effectively accurately predict traffic flow data with nonlinear relationships.

### Prediction results of support vector machine regression model

This article establishes a traffic flow prediction model based on the support vector machine regression algorithm and applies the model to actual road vehicle prediction. The predicted results are shown in Table 1.

As shown in Table 1, the number of vehicles in the morning and evening rush hours of a certain city section is analyzed by inputting data sequence numbers for different periods to predict the number of vehicles in that section the next day at the same time. Table 1 shows that the difference between the predicted and actual values is within five vehicles, with a slight difference. This indicates that the algorithm model can accurately predict road traffic flow.

**Table 1 Prediction results of support vector machine regression model.**

| Time period | Enter the data sequence number | Predicted value (vehicle) | True value (vehicle) | The difference between the predicted value and the true value (vehicle) |
|---|---|---|---|---|
| Morning peak | 10 | 789 | 785 | 4 |
| | 20 | 864 | 860 | 4 |
| | 30 | 675 | 676 | 1 |
| | 40 | 942 | 945 | 3 |
| | 50 | 854 | 857 | 3 |
| Evening peak | 10 | 635 | 638 | 3 |
| | 20 | 854 | 851 | 3 |
| | 30 | 742 | 740 | 2 |
| | 40 | 614 | 617 | 3 |
| | 50 | 358 | 354 | 4 |

The development of a traffic flow forecast model is highly significant. It can furnish insights on forthcoming traffic conditions, aiding traffic managers in the efficient planning and allocation of traffic resources. Secondly, traffic flow prediction algorithms can forecast future traffic conditions for drivers and travellers. Precise traffic flow forecasting can assist individuals in circumventing crowded road segments, selecting optimal routes, conserving travel time, and diminishing fuel consumption, thus enhancing the travel experience and navigation services. The traffic flow prediction model can forecast traffic bottlenecks and identify traffic abnormalities. The model identifies anomalous events and provides prompt alerts through the analysis of historical and real-time data. This can help reduce accident risks, improve traffic safety, and provide an essential reference for urban planners and traffic decision-makers. Establishing a traffic flow prediction model requires many historical and real-time traffic data, data science, and machine learning techniques for modeling and prediction. This promotes the gradual transformation of traffic management to data-driven and intelligent and encourages the development of urban traffic management towards intelligent transportation systems.

### Optimization scheduling algorithm
#### Constraints
Because vehicle scheduling is a complex process, genetic algorithms are prone to errors as the scheduling process becomes more complex (*Shengdong, Zhengxian & Yixiang, 2019*). Therefore, certain constraints need to be added. When solving the intelligent vehicle scheduling model in this article, there are no constraints on all conditions, as it is almost impossible to constrain all situations and only needs to determine whether the scheduling task chromosome is qualified. If qualified and placed in the next generation group, a special penalty factor can be set to reduce its related fitness if a certain constraint principle is exceeded. In this way, after several generations of genetic algorithm execution, some scheduling tasks that do not meet the constraint conditions are gradually eliminated. Only a relatively small portion would enter the model and would not dominate, which is

beneficial for calculating the optimal solution. When considering the constraint of road capacity, the corresponding penalty coefficient can be used to handle it, and the capacity factor of the road can be taken into account to form an objective function equation:

$$\min x = \sum_{o=1}^{m}\sum_{k=1}^{m}\sum_{l=1}^{m} v_{ko}c_{kol} + Z\sum_{l=1}^{e}\max\left(\sum_{o=1}^{m} h_o u_{ol} - w_l, 0\right). \tag{10}$$

$Z\sum_{l=1}^{e}\max\left(\sum_{o=1}^{m} h_o u_{ol} - w_l, 0\right)$ indicates that if the road capacity constraint is not met, it needs to be multiplied by the penalty value. In the algorithm, the constraint condition can be $Z \to \infty$. This is impossible in reality, it can choose a larger positive number.

In the process of traffic scheduling, some vehicles arrive at a fixed location at a certain time in advance, such as ambulances, and this situation needs to be judged, otherwise punishment must be imposed. Under the constraints of scheduling time, each vehicle has a reasonable scheduling time, assuming this time range $[s_k, n_k]$, where $s_k$ is the start time of scheduling task k. $n_k$ is the time when the scheduling task k completes the scheduling. This interval has strict restrictions, and all vehicles have their own interval. If this period is exceeded, the scheduling task needs to be delayed. If the scheduling task enters the scheduling cycle, there is the following relationship:

$$d_o = 0, s_k \le d_k \le n_k. \tag{11}$$

If f represents the loss of vehicles waiting for scheduling, r represents the additional penalty value required to complete a complete scheduling.

If the dispatching vehicle reaches the area that needs to be dispatched before $s_k$, the scheduling cost would be $f(s_k - d_k)$. If the dispatching vehicle reaches the dispatching area, an additional penalty of $r(d_k - n_k)$ would be imposed. The objective function of the vehicle scheduling model with time constraints obtained by using this time constrained penalty method is:

$$\begin{aligned}\min x = \sum_{o=1}^{m}\sum_{k=1}^{m}\sum_{l=1}^{m} v_{ko}c_{kol} + Z\sum_{l=1}^{e}\max\left(\sum_{o=1}^{m} h_o u_{ol} - w_l, 0\right) \\ + f\sum_{k=1}^{e}\max(s_k - d_k, 0) + r\sum_{k=1}^{e}\max(d_k - n_k, 0).\end{aligned} \tag{12}$$

*Solution*

After calculating reasonable constraint conditions, randomly generate n task sets that need to be scheduled, such as $o_1, o_2, \ldots, o_n$, if

$$\begin{aligned}\sum_{k=1}^{d-1} h_{ko} \le w_o \\ \sum_{k=1}^{d} h_{ko} > w_o.\end{aligned} \tag{13}$$

Move the genes from $d$ to n backwards one by one, the first position would be vacated, and the last position would be inserted into the first position. Then, if

$$\sum_{k=d}^{u-1} h_{ko} \leq w_o$$
$$\sum_{k=d}^{u} h_{ko} > w_o. \tag{14}$$

Repeat the above steps to leave the $u$ bit blank and insert 0 into position $u$.

In order to further optimize the encoding process, according to traditional encoding principles, scheduling tasks l can be randomly generated to form a unified scheduling population. Recorded as:

$$H_p = \{h_0, h_1, \ldots, o_l\}. \tag{15}$$

For the fusion of road conditions without saturation, establish a scheduling model, and the solution process is as follows:

① Set a scheduling initial standard variable $y_m = 0$ (m $= 1, 2, \ldots, t$) to determine whether a vehicle meets the conditions for scheduling.

② In a congested state, each vehicle that needs to be dispatched can form a loop. Assuming that there are $m_r$ vehicles that need to be scheduled in the r th path that can be scheduled, where r $= 1, 2, \ldots, T$. Let $Y_r$ be the set of scheduled vehicles included in the r path, with $Y_r = \varnothing$ in the starting state. Assuming that the scheduling importance of each vehicle is $B_r$ and a copy of $B_r$ is retained, $B'_r = B_r$ is assumed. The customer point corresponding to the a position in the r sub circuit is marked as $t_{ra}$, with an initial value of 0. The initial value of the iteration variable j is 1. Perform iterative scheduling calculations. During the calculation process, at the j th iteration, there is a correlation between the scheduling vehicle c and the sub loop r:

$$c = \sigma_j - \left\lceil \frac{\sigma_j - 1}{\mu} \right\rceil \mu \tag{16}$$

$$r = \left\lceil \frac{\sigma_j - 1}{\mu} \right\rceil + 1. \tag{17}$$

③ Detect the scheduling flag variable $r_c$ of the vehicle. If this value is equal to 0, then the vehicle has not been fully scheduled and needs to be scheduled. Otherwise, proceed to step ⑤.

④ If the requirement $g_c$ of vehicle c for scheduling is less than the road capacity $B'_r$ of the r-th vehicle in the sub loop, then $r_c = 1, B'_r = B_r - g_c, m_r = m_r + 1 = 1$. If it is not equal, proceed to step ⑤.

⑤ Conduct a new round of iterative calculation. If j equals T $\times$ $\mu$ + 1, it indicates that T $\times$ $\mu$ elements in the chromosome have been detected and all vehicles have completed effective scheduling in congested conditions. If this condition is true, it indicates that the vehicles have already been allocated in the sub loop, and the path set of the sub loop is R$\varphi = \{R_1, R_2, \ldots, R_r\}$. The obtained solution is a feasible method for completing a

congestion scheduling. This method needs to be further validated to determine if it is optimal. If it is not optimal, proceed to step ③ to continue execution.

*Comparison of results before and after scheduling*

This article uses optimization scheduling algorithms to optimize the scheduling of congested roads, and compares the number of stops on frequently congested road sections, average waiting time of vehicles, and average speed of vehicles before and after the application of scheduling algorithms, as shown in Fig. 3.

As shown in Fig. 3, this article analyzes the number of stops, average waiting time, and average speed of vehicles on frequently congested road sections in a certain city before and after the application of scheduling algorithms at the same time. Figure 3A shows the number of stops on frequently congested sections before and after using the scheduling algorithm. The abscissa is the time, in seconds, and the ordinate is the number of stops. It can be clearly seen from Fig. 3A that the number of stops on this road section fluctuated around 60 before using the scheduling algorithm, while the number of stops after using the scheduling algorithm fluctuated around 30. It indicates that effective scheduling of frequently congested road sections through scheduling algorithms greatly reduces the number of stops for vehicles on that road section.

Figure 3B shows the average waiting time of vehicles on frequently congested road sections before and after the use of scheduling algorithms. This statistics shows the waiting time for the red light on the road section. From Fig. 3B, it can be clearly seen that the waiting time for vehicles at the red light before applying the scheduling algorithm fluctuated around 75 s. After applying the scheduling algorithm, the time for vehicles to wait for red lights fluctuated around 65 s, indicating that the application of the scheduling algorithm greatly shortens the time for vehicles to wait for red lights on congested road sections.

Figure 3C shows the average speed of vehicles on frequently congested road sections before and after the use of scheduling algorithms, which measures the speed at which vehicles are traveling on that road section. The slower the congested vehicles travel, the faster the less congested vehicles travel. From Fig. 3C, it can be seen that the driving speed of vehicles after applying the scheduling algorithm is faster than that before applying the scheduling algorithm, indicating that the effective scheduling algorithm ensures the smoothness of the road, thereby improving the driving speed of vehicles on this section. By optimizing route planning, reducing congestion, improving transportation efficiency, saving resources and energy, and providing more efficient and sustainable solutions for logistics transportation and traffic management, vehicle scheduling algorithms have the potential to create solutions that are more efficient and sustainable (*Saleem et al., 2022*). It is of great significance to improve the efficiency of the transportation system, reduce logistics costs, and improve environmental quality.

## Intelligent traffic signal control

This article utilizes artificial intelligence algorithms to optimize urban traffic signal control systems, dynamically adjust signal timing based on real-time traffic conditions, reduce traffic congestion and delays, and improve road traffic efficiency.

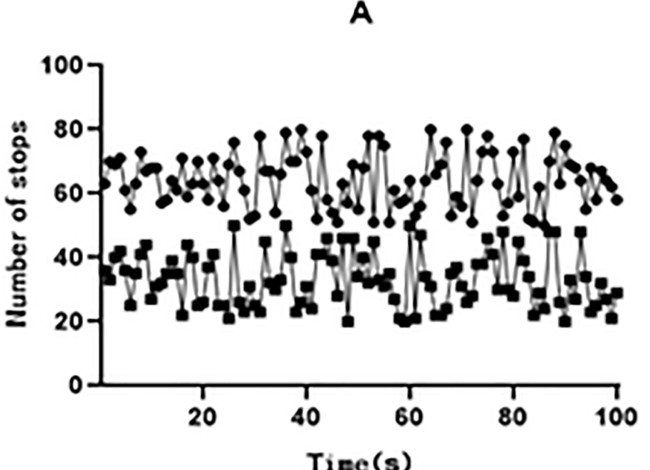

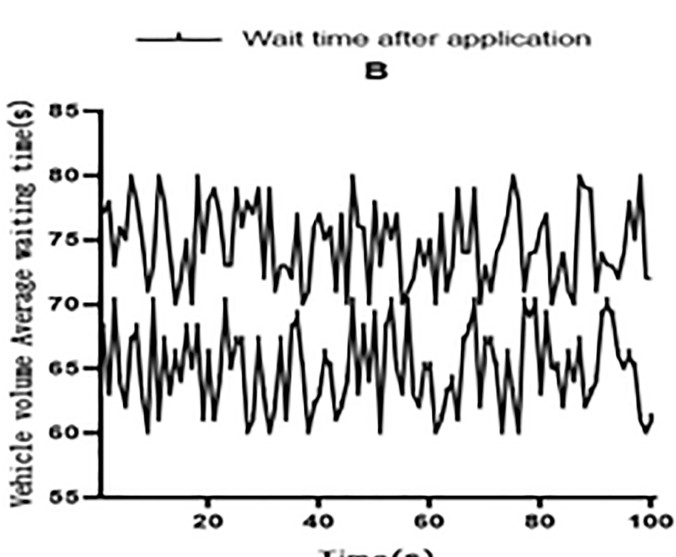

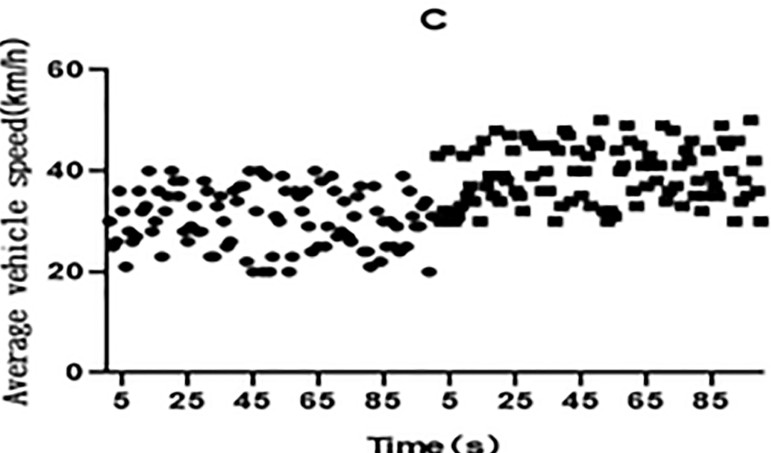

**Figure 3 Comparison of results before and after the application of scheduling algorithms on frequently congested road sections in a certain city.**

### Mathematical model of urban traffic signal control

The current urban traffic signal control time has a certain delay (*Jinsong & Mingjun, 2020*). After long-term research and summary by many experts, the average delay time f calculation formula is proposed as follows:

$$f = \frac{V(1-\varepsilon)^2}{2(1-\varepsilon c)} + \frac{c^2}{2w(1-c)} - 0.65\left(\frac{V}{w^2}\right)\frac{1}{3}c^{(2+5\varepsilon)}. \tag{18}$$

Among them, c is the saturation of traffic flow; V is the time period of the signal light; $\varepsilon$ is the time of the green signal light; w is the traffic volume (pch/h).

In practical applications, it was found that the value of $0.65\left(\dfrac{V}{w^2}\right)\dfrac{1}{3}c^{(2+5\varepsilon)}$ in the formula is very small, so it can be ignored in actual calculations to obtain the calculation formula for the actual average delay time f:

$$f = \frac{V(1-\varepsilon)^2}{2(1-\varepsilon c)} + \frac{c^2}{2w(1-c)}. \tag{19}$$

Taking the intersection of a real city as an example, the total delay time at that intersection can be calculated using the following formula:

$$F = \sum_{o=1}^{4}\sum_{k=1}^{2}\left\{w_{ko}\left[\frac{V(1-\varepsilon_o)^2}{2(1-\varepsilon_o c_{ko})} + \frac{c_{ko}{}^2}{2w_{ko}(1-c_{ko})}\right]\right\}. \tag{20}$$

Among them, $w_{ko}$ is the traffic flow at the k entrance and exit in the direction allowed by Class o signal lights. $c_{ko}$ is the saturation of traffic flow at the entrance and exit of the o class signal lamp and the k class signal lamp. $\varepsilon_o$ is the time allowed for the green signal light in the direction of travel for Class o signal lights, $\varepsilon_o = y_o/V$.

As shown in Formula (21), assuming the time of the green signal light is $r$, the time of the green signal light needs to meet the following constraints:

$$y_o \le r \le V - A - 30. \tag{21}$$

Among them, $A$ is the time lost due to delay at the intersection.

Furthermore, assuming that the maximum saturation value of the traffic flow at the intersection does not exceed 0.8, then $y_o$ needs to meet the following constraint conditions:

$$y_o = h_{ro} \ge Vy_{o,max}/0.8. \tag{22}$$

Among them, $y_{o,max}$ is the maximum flow ratio at the intersection, and $h_{ro}$ is the effective time.

In summary, the set of constraints for the urban traffic signal control model is:

$$\begin{cases} \sum_{o}^{4} y_o = V - A \\ y_o \le r \le V - A - 30 \\ y_o = h_{ro} \ge Vy_{o,max}/0.8. \end{cases} \tag{23}$$

### Signal lamp controls four routes of vehicle driving

This article uses artificial intelligence technology to establish a mathematical model for urban traffic signal control, and analyzes the four driving routes of vehicles controlled by urban traffic signals based on the model, as shown in Fig. 4.

As shown in Fig. 4, the left side is a schematic diagram of an intersection, and the lines with arrows in different colors at the intersection are the schematic diagrams of four routes where vehicles are controlled by signal lights. These four routes are represented by A, B, C, and D, as shown in the upper right corner of Fig. 4. The directions of the entrance and exit of this intersection are up north, down south, left west, and right east, with specific markings shown in the bottom right corner of Fig. 4. Route A has four directions of travel. The first type is from west to east, the second type is from west to south, the third type is from east to west, and the fourth type is from east to north. Route B has two directions of travel, the first from west to north, and the second from east to south. Route C has four directions of travel. The first type is from north to south; the second type is from north to east; The third type is from south to north; the fourth type is from south to west. Route D has two directions of travel; the first is from north to west; the second type is from south to east. By controlling signals and allocating vehicles appropriately, it is possible to ensure residents' safety while effectively alleviating traffic pressure and avoiding traffic accidents that can lead to road congestion.

### Using models to calculate flow ratio

The urban traffic signal control model established above is used to calculate the traffic flow at intersections in the direction consistent with Fig. 4, as shown in Table 2.

As shown in Table 2, assuming the time period of the signal light is 130 s, the lost time due to delay is 10 s, and the minimum green signal time of the signal light at the intersection is 10 s. Assuming that the time of the yellow signal light is the same as the time lost due to delay, then the actual time of the green signal light shall prevail.

Taking the route allowed by Class A signal lights as an example, taking a relatively large $y$ value and substituting it into Formula (22) can obtain $y_A \geq 34$. Similarly, it can be concluded that Class B signal lights allow a flow ratio of $y_B \geq 29$ on the driving route; Class C signal lights allow a flow ratio of $y_C \geq 34$ on the driving route; Class D signal lights allow a traffic ratio of $y_D \geq 22$ on the driving route.

### Intelligent driving and traffic safety

This article applies artificial intelligence technology to develop an intelligent driving system, including autonomous driving technology, intelligent traffic monitoring, and safety warning, to improve traffic safety and driving efficiency, as shown in Fig. 5.

As shown in Fig. 5, an intelligent driving system is established using artificial intelligence technology. The intelligent driving system utilizes sensors and algorithms to perceive the surrounding environment of the vehicle in real-time, providing automated driving assistance functions (*Jinsong & Mingjun, 2020*; *Manoharan, 2019*). Through accurate perception of road conditions, obstacles and other vehicles, human error and driving errors can be reduced, thereby reducing the risk of traffic accidents and improving road safety.

AI technology makes use of current data coming from sensors, cameras and other connected vehicles to avoid or reduce any mishap. With regard to route surveillance, AI is capable of identifying threats related to vehicle speed or distance to other vehicles on the

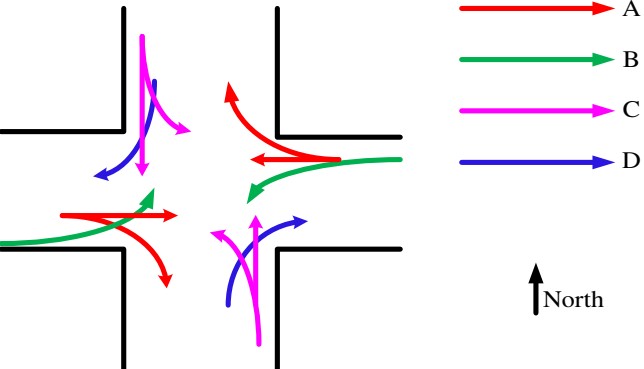

**Figure 4 Diagram of the driving route of vehicles with four types of signal lights.**

**Table 2 Calculation of traffic flow in each direction at an intersection.**

| Traffic lights control the type of route the vehicle is traveling | Entrance and exit directions | Traffic flow | Saturation of traffic flow | Flow ratio |
|---|---|---|---|---|
| A | East | 352 | 2,000 | 0.176 |
|   | West | 424 | 2,000 | 0.212 |
| B | East | 162 | 900 | 0.18 |
|   | West | 135 | 900 | 0.15 |
| C | South | 315 | 1,800 | 0.175 |
|   | North | 387 | 1,800 | 0.215 |
| D | South | 126 | 900 | 0.14 |
|   | North | 108 | 900 | 0.12 |

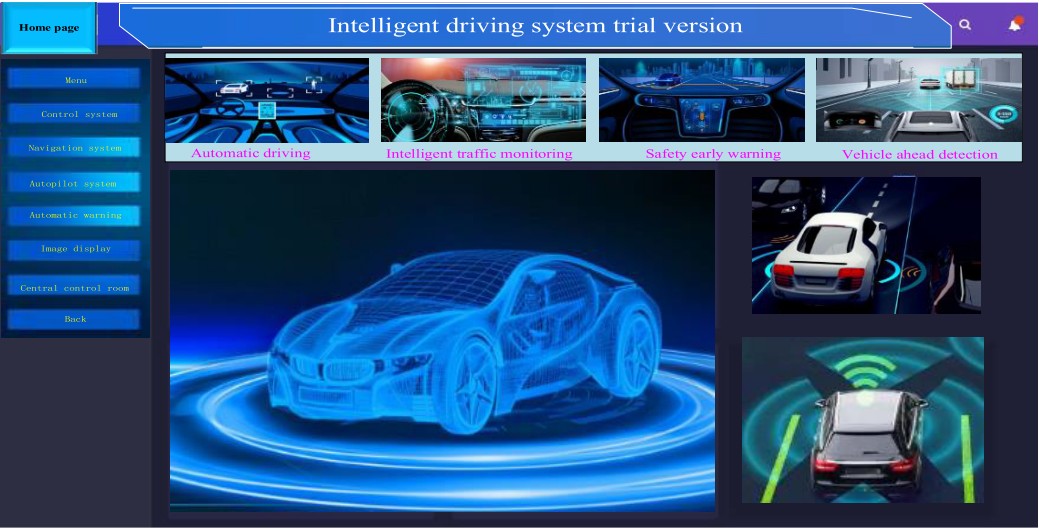

**Figure 5 Interface diagram of intelligent driving system.**

road, or road conditions, and thereafter, warning the driver. For example, AI systems can suggest new driving patterns, such as much slower speeds, larger buffer distances, or staying off dangerous roads in harsh weather. Also, the aid systems, for instance, lane departure and autonomous emergency brake, assist in increasing safety provision since they take over the wheel during adverse circumstances. These kinds of forecasts are not only used in the prevention of accidents that sooner or later occur but also in training safer driving behaviours to minimize the possibility of repeat mishaps. Whenever new information is incorporated from previous and current situations, an AI system can change its design according to the different possibilities, thereby improving safety and reliability of transportation in cities.

The intelligent driving system has the ability to optimize vehicle path selection and traffic scheduling. In short, with the continuous progress and promotion of technology, intelligent driving systems would play an increasingly important role in the future transportation field (*Vangala et al., 2020*).

AI systems in urban intelligent transportation are characterized by fast responses due to real-time data from sensors, cameras, and other connected vehicles, allowing for quick reaction to changes in flow. When they discern changes like an accident or an abrupt disruption due to weather conditions, these systems evaluate the information within seconds. They can alter the traffic signals to allow free-flowing traffic or caution other drivers through map applications to avoid congested areas. For instance, if there is an accident on the road, the AI directs pathways for use by emergency vehicles and simultaneously directs other pathways not accessible by the affected vehicle. When it rains, AI can estimate that my travel may be interfered with, and I shall be given an early warning with better routes to take. This capability is achieved using machine learning models that have historical and real-time data feeds as inputs to the system, which increases the system's ability to handle conditions that are unforeseen on the fly. Real-time analysis ensures tutored and efficient traffic flow and safety enhancement within express cities.

## Data analysis and decision support

After the above analysis, this article utilizes artificial intelligence technology to analyze a large amount of traffic data, providing decision support for urban traffic managers. Firstly, it utilizes artificial intelligence technology to mine and analyze a large amount of traffic data, discovering hidden patterns, trends, and patterns in the data. Secondly, artificial intelligence technology can be combined with sensors, cameras, and other monitoring devices to collect and process traffic data in real-time, which helps to respond to traffic problems in a timely manner, optimize traffic mobility, and take timely measures to improve road safety. In addition, artificial intelligence technology can be used to establish complex models and optimize traffic data, in order to find the best traffic scheduling and optimization strategies. Finally, the traffic data is visualized using AI technology. Through charts, maps and data and information visualization interfaces, urban traffic managers can understand the traffic conditions and problems more intuitively. Through this visual analysis method, decision-makers can better understand traffic data and make decisions and plans based on the data, thereby improving the effectiveness of traffic management

(*Caiyun, Yongsheng & Ge, 2021*). Overall, it can effectively provide decision support by utilizing artificial intelligence technology to analyze a large amount of traffic data. This can help urban traffic managers better monitor and manage traffic conditions, promoting smooth, safe, and sustainable development of traffic.

## Public engagement in AI implementation

Infomation sharing is one of the most important parts of building AI systems for managing traffic. It also explains the role of residents in the implementation process through sensitisation and participation through feedback and consultation. Public awareness campaigns can inform residents why self-driving cars are great for the population as a whole by making travel shorter, safer and, in general, less polluting. Residents can also give their opinions through questionnaires and open forums to ensure that the feedback given meets the needs of the system. Moreover, pilot testing that targets certain areas enables the implementation to be done in phases, and also the society adapts to the technology. As the consequence of following the approach that directly tackles interest and engagement of the general public, this model allows for more effective implementation of AI technologies and better long-term performance of urban traffic management systems.

## ECONOMIC BENEFIT EVALUATION

### Data sources

In traffic analysis for this study, the collected data from cross-sectional surveys from various regions to gain the best insights. These are real-time data from sensors deployed in cities across intersections, GPS data from municipal transport bodies, and traffic archive data from existing urban traffic information systems. Moreover, video feeds from the surveillance cameras as well as data from IoT devices generated rich traffic information regarding movement and flow, congestion, and events and incidents. These datasets were augmented with environmental data, including weather information from the local meteorological offices. The integrated data analysis increases the confidence and stability of the conclusions to AI effects on urban transport systems. Evaluating the investment cost and benefit of artificial intelligence technology in urban intelligent transportation systems can also detect the advantages of urban intelligent transportation systems. This article takes three congested roads of different degrees in a city as the experimental object, makes an economic evaluation of the time cost and labor cost of congested roads before and after the application of AI technology in an urban intelligent transportation system, records the data and analyzes the experimental results.

### Data analysis

#### Time cost

This article applies artificial intelligence technology to the urban intelligent transportation system in three congested city sections at different levels. It carries out time cost statistics from four aspects: waiting time for red lights, time for traffic congestion, time for accident resolution, and time ultimately required to pass the sections. It is assumed that the

economic cost of urban intelligent transportation systems before and after the application of AI technology is calculated at 10 yuan per minute, as shown in Table 3.

As shown in Table 3, the total time before the application of artificial intelligence technology in three congested sections of the city was between 100–200 min, and the calculated economic cost was 1,000–2,000 yuan. After applying artificial intelligence technology to three congested sections of the city with varying degrees of congestion, it ranged from 50 to 120 min. The calculated economic cost ranged from 500 to 1,200 yuan, indicating a significant improvement in time cost for three congested road sections. The time cost is relatively low, which can significantly reduce car fuel consumption and travel costs.

### Labor costs

This article applies artificial intelligence technology to three different levels of congested road sections in the city and calculates labor costs. Labor cost refers to the use of a large number of manual means to command and allocate congested road sections. Labor costs are mainly calculated based on the number of people commanding traffic, the number of people handling traffic congestion, and the number of people solving accidents. This article assumes that 50 yuan per person is used to calculate the artificial economic cost before and after the application of AI technology in the urban intelligent transportation system, as shown in Table 4. Although application of traffic systems greatly lowers the labor expenses through deployment of intelligent traffic systems, this has an aspect of foe within acquaintance general traffic systems since the development itself bears an adverse impact within the worker genetically occupied within general traffic systems. Staff performing manual traffic control and congestion may find themselves displaced from their positions. However, the shift to new, AI-driven systems is also a chance for workforce reskilling for upskilling. Sectors seeking to reduce threats of job loss as a result of AI technology adoption can involve the development of systematic training sessions to enable workers displaced to master specific tasks involving AI system management, maintenance, and data analysis. Further, new employment opportunities may be developed and connected with new technologies in development, integration and newly planned cities' systems, so there will always be needed personnel to work with intelligent transport systems.

As shown in Table 4, the economic cost of applying artificial intelligence technology to three different levels of congested road sections in the city was above 300 yuan. In contrast, the economic cost of applying artificial intelligence technology was reduced to below 300 yuan. Therefore, it can be effectively evaluated that the application of AI technology in urban intelligent transportation systems can greatly reduce labor costs, thus reducing road finance expenditure. The saved funds can be used to promote the development of other aspects of the city.

The new AI-driven systems of traffic maintenance were then compared with traditional traffic management methods to evaluate the number of economic gains in terms of efficiency and cost. The time costs of conventional methods were 112.5–194.4 min overall, and the financial costs were 1,125–1,944 yuan (see Table 3). When integrated with AI, the time costs were 59–110 min, and the cost significantly decreased to 590–1,100 yuan.

**Table 3 Time cost of artificial intelligence technology in urban intelligent transportation system.**

| | The congestion level of the road | Waiting time at the red light (minutes) | Traffic jam time (minutes) | Incident resolution time (minutes) | Time required to complete the road section (minutes) | Total time (minutes) | Economic cost (yuan) |
|---|---|---|---|---|---|---|---|
| Pre-application technology | Jam | 2.5 | 60 | 40 | 10 | 112.5 | 1,125 |
| | Relatively congested | 3.2 | 80 | 60 | 15 | 158.2 | 1,582 |
| | Very congested | 4.4 | 90 | 80 | 20 | 194.4 | 1,944 |
| After application of technology | Jam | 2 | 30 | 20 | 7 | 59 | 590 |
| | Relatively congested | 3 | 40 | 30 | 12 | 85 | 850 |
| | Very congested | 4 | 50 | 40 | 16 | 110 | 1,100 |

**Table 4 Labor cost of AI technology in the urban intelligent transportation system.**

| | The congestion level of the road | The number of people directing traffic | Number of people to deal with in traffic jams | The number of people who resolved the accident | Headcount | Economic cost (yuan) |
|---|---|---|---|---|---|---|
| Pre-application technology | Jam | 2 | 2 | 3 | 7 | 350 |
| | Relatively congested | 2 | 3 | 4 | 9 | 450 |
| | Very congested | 3 | 4 | 5 | 12 | 600 |
| After application of technology | Jam | 0 | 1 | 2 | 3 | 150 |
| | Relatively congested | 0 | 1 | 2 | 3 | 150 |
| | Very congested | 0 | 2 | 3 | 5 | 250 |

Labour costs were also reduced; in the conventional system, human interfaces deployed 350–600 yuan, whereas AI-integrated interfaces optimized the amount to 150–250 yuan with less human intervention (Table 4). In aggregate, the cost-cutting model of employing AI was 6,051 yuan under the traditional system, whereas its AI counterparts were 3,090 yuan, thereby showing an overall saving of 2,961 yuan. These results support the argument that AI plays a significant part in enhancing cities' traffic flow and other resource efficiency.

*Comparison of economic costs*

This article comprehensively compares the time economic and artificial economic costs in Tables 3 and 4 to better evaluate the economic benefits of urban intelligent transportation systems after the application of artificial intelligence technology, as shown in Fig. 6.

As shown in Fig. 6, it is evident that both the time and labor economic costs before the application of AI technology in urban intelligent transportation systems are much higher than those after the application of AI technology in urban intelligent transportation systems. The total economic cost before applying artificial intelligence technology was 6,051 yuan; the total economic cost after applying artificial intelligence technology was

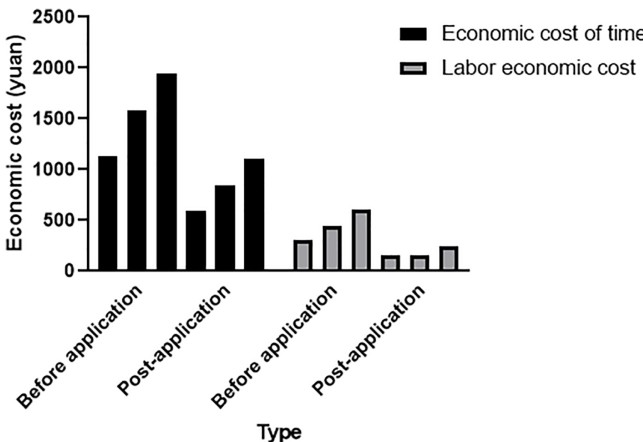

**Figure 6** **Comparison of time economic cost and labor economic cost.**

3,090 yuan; the total economic cost after applying artificial intelligence technology was 2,961 yuan less than the total economic cost before applying artificial intelligence technology, which significantly reduces the maintenance cost of roads. AI outperforms traditional methods in processing real-time data, enabling quicker and more accurate decision-making in dynamic environments such as urban transportation systems. Its ability to continuously adapt and learn from vast datasets makes it more scalable for large and complex transportation networks, unlike traditional methods that struggle with such systems. Additionally, machine learning models used in AI systems offer higher precision in predicting and optimizing traffic flows compared to static algorithms used in traditional methods.

# DISCUSSION AND OUTLOOK

## Economic growth

Artificial intelligence technology in urban intelligent transportation systems can positively impact urban economic growth. Firstly, it is to improve traffic efficiency through the precise prediction, real-time monitoring, and optimized scheduling capabilities of artificial intelligence technology. Intelligent transportation systems can reduce traffic congestion and enhance vehicle traffic efficiency. This helps to minimize transportation time costs and energy consumption, improve logistics and transportation efficiency, and promote the smooth operation of the urban economy. The second is to reduce the transportation cost. Artificial intelligence technology in the intelligent transportation system can help urban planners and managers more accurately adjust the transportation facilities and infrastructure to optimize the road network layout and traffic signal control. This can save on infrastructure construction and maintenance costs and reduce vehicle operating costs, providing cheaper transportation services for urban enterprises and residents. Next is to promote business activities and innovation, and intelligent transportation systems provide reliable traffic information and navigation services. This can make finding their destination more accessible, reducing the cost of commercial activities and personnel mobility. At the

same time, the development of intelligent transportation systems has also spawned a series of innovative enterprises and employment opportunities involving data analysis, traffic management, vehicle-to-everything and other fields, promoting the diversified development of the urban economy. More importantly, it can improve the quality of the urban environment. The application of artificial intelligence technology promotes the transportation system to be more efficient and green. By optimizing traffic flow and reducing vehicle congestion, intelligent transportation systems can reduce exhaust emissions, improve air quality and reduce environmental pollution. This helps improve the quality of life in cities, attract more talents and investment, and promote sustainable economic development.

## Data privacy and protection

Using AI in transportation requires data privacy and thus holds significant importance in its application. These systems utilize large volumes of real-time information on vehicle and passenger movements, possibly personal data. Data security shall be maintained by adopting strict measures for data protection, such as the use of encryption, anonymization of data, and any access to data. This guarantees the public's trust in AI and makes the collection transparent and legal. When privacy concerns are solved, it will protect an individual's rights to privacy and help achieve acceptance of AI technologies in urban transport systems.

## Sustainable development potential

The application of artificial intelligence technology in urban intelligent transportation systems has broad potential and effect on sustainable development. The first is intelligent transportation management and optimization. Artificial intelligence technology can establish an intelligent traffic management system to achieve automated traffic scheduling and optimization. This would bring benefits such as efficient use of traffic resources, relief of road congestion, intelligent control of traffic signals, *etc.*, thus promoting the sustainable development of urban traffic systems. Next are energy conservation, emission reduction, and environmental protection. Artificial intelligence technology in urban intelligent transportation systems can improve energy utilization efficiency and reduce vehicle stagnation and idle time by optimizing traffic flow and traffic light timing, thus reducing exhaust emissions and energy consumption. This helps improve air quality, reduce carbon emissions, and promote sustainable urban environment protection. The next step is to reduce traffic accidents and improve traffic safety. Artificial intelligence technology can conduct real-time traffic monitoring and early warning in the intelligent transportation system, help identify traffic accident risks, and take timely measures to avoid accidents. Through intelligent traffic supervision and driving assistance systems, the incidence of traffic accidents can be reduced, the safety of traffic participants can be ensured, and the sustainable safety of cities can be improved. Then, there are personalized travel and multimodal transportation. The application of artificial intelligence technology in the urban intelligent transportation system can provide customized travel recommendations and planning for residents and traffic participants, making travel more efficient,

convenient and comfortable. At the same time, the promotion and integration of multimodal transportation, such as orderly coordination of public transportation, shared transportation, and nonmotorized transportation, can also promote the sustainable development of urban transportation.

End-user application of AI in smart cities with a specific focus on transportation systems is evidence for sustainable development. For example, Singapore's Intelligent Transport System uses AI to control traffic by allowing real-time analysis and prediction, cutting short on average vehicle idling time accompanied by emissions. But the AI congestion pricing model in Stockholm, for example, has been proven to be responding to the challenge by cutting traffic by 20%, hence cutting the rate of emission of CO2 and air pollution as well. Another excellent example is the application of AI for adaptive traffic signal control in Los Angeles, which has actually increased traffic/congestion by 10% of the average time during rush hour. These cases show how and when AI can be harnessed in different urban scenarios in order to support sustainability by reducing negative effects on the environment as well as improving transportation networks. Such implementations act as examples that cities that wish to integrate AI for sustainable development in the long run need to consider.

Finally, there is decision-making and planning based on data. AI technology can use a large amount of traffic data to conduct accurate analysis and prediction, help decision-makers formulate reasonable transportation planning and policies, and achieve sustainable urban traffic development. Data-driven decision-making and planning can help optimize the layout of transportation networks, improve transportation infrastructure, and provide more targeted traffic management solutions.

## Scalability of AI solutions

The ability of AI solutions to integrate ITS for various cities largely depends on the compatibility of the solutions with the environments of large and small towns as well as the flow of traffic. The proposed solutions fall in this category because the traffic flow prediction models and optimization algorithms proposed to address the traffic flow can be made to have variable system capabilities depending on the infrastructure and data capacity available on the ground. For the traffic networks of lower complexity, it is possible to work with the smaller-scale AI models, where the computational complexity is decreasing and AI is used only for important junctions. However, for large-scale urban environments with emerging and more complex traffic characteristics, the base models can be augmented with additional data feeds, such as IoT devices and GPS from car satnavs, to handle the higher throughput levels. Moreover, the structure of AI systems can be gradually implemented in stages. In contrast, an unlimited budget or infrastructure for the foundation is not required when deploying all the features, starting from signal optimization to congestion prediction. In addition, the present algorithms are also scalable to traffic fluctuations since the system can rely on real-time data analysis to detect situations of high traffic density during peak hours and low traffic density during other times. The following capabilities highlight the fact that the proposed solutions are general,

and this should make them relevant to much utility and effectiveness across most urban environments:

### AI in long term urban planning

AI can greatly impact long-term planning in cities by providing information on what forms of infrastructure should be built. Using traffic data and traffic distribution, population density, and commuters 'daily routines, AI systems can pinpoint which areas need new roads or require updating of existing structures and layouts. For example, predictors can predict traffic congestion patterns in the future to help urban planners avoid optically designing road networks to handle congestion. AI also helps to solve such automation challenges as the identification of passenger demand and the decisions on where and how to enhance or contract transportation offerings. Moreover, using environmental data, AI can suggest what infrastructure improvements should be considered as environmentally friendly as possible, such as bike paths, pedestrian walkways, or charging stations for electric vehicles. The integration of artificial intelligence means not only improving the existing decision-making process but also creating a flexible basis for future urban development. This will have the added effect of making cities more ready to grow in the future, make efficient use of resources, and reach for sustainability.

In conclusion, the application of artificial intelligence technology in urban intelligent transportation system has the potential and effect of sustainable development. It can improve the transportation system's efficiency, protect the environment, enhance traffic safety, and provide urban residents with a more intelligent and personalized travel experience. However, in promoting the sustainable development of artificial intelligence technology in urban transportation, attention needs to be paid to issues such as data privacy, security, and fairness, and corresponding regulations and measures should be taken to ensure its sustainability and social benefits.

## CONCLUSIONS

The application of artificial intelligence technology in urban intelligent transportation systems has far-reaching significance for economic development. It improves transportation and resource utilization efficiency, promotes commercial activities and innovation, and enhances the city's image and attractiveness. This can support planning and decision-making while contributing to cost savings and sustainable development. This article showed the advantages of artificial intelligence technology by analyzing the application of artificial intelligence technology in urban intelligent transportation systems and, on this basis, analyzed the economic benefits of roads after the application of artificial intelligence technology for evaluation. It also showed the sustainable development potential and effect of artificial intelligence technology in urban intelligent transportation systems, which can provide favorable support for promoting the prosperity and development of the urban economy. Labour cost savings achieved through AI-driven traffic systems can be reinvested in urban planning and infrastructure, such as redesigning road networks and enhancing public transit systems. This strategic allocation ensures long-term economic and social benefits, fostering sustainable urban development. To

facilitate the adoption of AI-based traffic systems, cities must address challenges such as high initial costs, integration with legacy infrastructure, and public acceptance. Phased implementation, leveraging public-private partnerships, and robust data privacy frameworks can mitigate these issues. Pilot projects and transparent communication with stakeholders will further enable smoother transitions, ensuring AI's potential is fully realized for sustainable urban growth.

### Funding
The authors received no funding for this work.

### Competing Interests
The authors declare that they have no competing interests.

### Author Contributions
- Ziming Zhao conceived and designed the experiments, performed the experiments, analyzed the data, performed the computation work, prepared figures and/or tables, and approved the final draft.
- Jinyu Chen conceived and designed the experiments, performed the experiments, analyzed the data, performed the computation work, prepared figures and/or tables, authored or reviewed drafts of the article, and approved the final draft.

### Data Availability
The code is available in the Supplemental File.

The data is available at GitHub and Zenodo:

- https://github.com/weisongwen/UrbanLoco/tree/master

- Ziming, Z. (2025). Application of Artificial Intelligence Technology in the Economic Development of Urban Intelligent Transportation System [Data set]. Zenodo. https://doi.org/10.5281/zenodo.14882363.

### Supplemental Information
Supplemental information for this article can be found online at http://dx.doi.org/10.7717/peerj-cs.2728#supplemental-information.

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
