# Peer review of "Application of artificial intelligence technology in the economic development of urban intelligent transportation system"

_PeerJ Computer Science, doi:10.7717/peerj-cs.2728_

## Round 0.1 · original submission · Major Revisions

Please see a detailed review of both reviewers. Suggestions include expanding the literature review, addressing public engagement, scalability, and long-term urban planning, and clarifying datasets, algorithms, and infrastructure needs. Recommendations also emphasize improving figures, addressing grammatical issues, and exploring data privacy and real-time traffic adjustments. These areas will strengthen the paper's clarity, credibility, and impact.

Reviewer 1 ·

Basic reporting

This paper provides an insightful analysis of integrating artificial intelligence (AI) into urban intelligent transportation systems (ITS), presenting its economic, traffic, and environmental benefits. The paper highlights the transformative potential of AI in improving traffic flow, reducing congestion, optimizing signal control, and contributing to sustainable urban development. The use of AI for economic evaluation and sustainability in transportation systems is a timely and important subject, and your methodology and data analysis are well executed. Overall, the paper offers valuable insights into how AI can be leveraged to address key urban traffic challenges. However, the paper could be strengthened in certain areas. Below are some suggested revisions and comments that can help improve it.
* The paper has a solid foundation, but the literature review could be expanded. A separate section is recommended after the introduction to highlight recent advancements in AI and urban ITS. This will provide a clearer context and show how the study builds on existing research.
* The economic analysis is strong, but comparing it with traditional traffic management methods (before AI implementation) would provide a clearer baseline for understanding AI's improvements
* The impact of AI on long-term urban planning is not addressed. Discussing how AI can inform future infrastructure decisions, such as new road layouts or public transportation needs, would be helpful.
* The discussion of intelligent traffic management is insightful, but it could be expanded to include the integration of AI with existing urban infrastructure. For example, how would AI systems interact with current traffic light management or vehicle detection systems?
* The figure labeling is unclear in places (e.g., Figure 4 and Figure 5). The figures could be more descriptive, and the axes or elements within them should be better explained in the captions to avoid confusion.
* The paper contains quite a few grammatical errors that need attention. A careful review to correct these issues would improve the overall readability and professionalism of the work.

Experimental design

The methodology section mentions traffic flow models and AI integration but does not clearly explain the specific algorithms used. Including a brief explanation of why particular algorithms (e.g., SVM, Decision Trees) were chosen over others could improve clarity.
Data privacy is a crucial issue when discussing AI applications. The paper could benefit from briefly mentioning how personal data will be handled and protected in AI-driven transportation systems.

Validity of the findings

The paper does not discuss scalability—how easily can the proposed AI solutions be applied to cities of different sizes or with varying traffic patterns?

Additional comments

The paper could provide more detail on real-time traffic adjustments. How quickly can AI respond to changes in traffic conditions, such as accidents or weather disruptions?

Reviewer 2 ·

Basic reporting

1. The sustainability discussion is strong, but it would be valuable to include specific examples of cities or projects where AI has already been successfully implemented to reduce emissions and improve traffic efficiency.
2. There is no mention of public engagement. How will residents be involved in the transition to AI-driven traffic systems? Public perception and acceptance could play a crucial role in the success of such systems.
3. The AI-driven traffic monitoring section would benefit from discussing the hardware and infrastructure requirements for deploying AI at scale, including sensors, cameras, and communication networks.
4. The data sources used for traffic analysis should be clarified. Are these datasets obtained from city-wide sensors, vehicle GPS data, or something else? A more detailed explanation would enhance the study's credibility.

Experimental design

5. The conclusion could include more actionable recommendations for cities adopting AI-based traffic systems. What are the main challenges cities face in implementation, and how can they be overcome?
6. The discussion of labour cost reduction is relevant but could also address the potential impact on employment in traffic management sectors. What are the implications for workers in traditional traffic roles?
7. Figures 4 and 5 should be improved in terms of clarity. For example, the directions in Figure 4 could be labeled more explicitly, and a more detailed explanation of how these routes align with traffic flow would improve understanding.

Validity of the findings

8. The AI technology section mentions its potential to reduce accidents, but it would be beneficial to explore how AI can predict and prevent accidents based on real-time data. For example, could AI recommend changes to driving behaviour?
9. Labour cost savings are noted, but discussing how the savings will be reinvested into other areas (e.g., urban planning, infrastructure) would add depth to the paper’s conclusion.
10. The discussion on AI in urban traffic could explore how these technologies might evolve as part of smart city initiatives in the future, integrating seamlessly with other systems like energy grids or water management.
11. There are several grammatical errors throughout the paper. I suggest a thorough review to correct these issues and improve clarity and coherence. Ensuring proper grammar and sentence structure will enhance the overall quality of the work.

---

## Round 0.2 · Major Revisions

Please see both detailed reviews. Both reviews commend the paper for its insightful analysis of AI integration into urban intelligent transportation systems, highlighting its economic, traffic, and environmental benefits. However, they suggest several improvements: expanding the literature review, comparing AI with traditional methods, addressing long-term urban planning, clarifying figure labels, and correcting grammatical errors. Additionally, they recommend discussing scalability, data privacy, real-time traffic adjustments, public engagement, hardware requirements, and the impact on employment. The reviews also emphasize the need for more detailed explanations of algorithms, data sources, and actionable recommendations for cities adopting AI-based systems. Overall, while the paper is strong, these enhancements would strengthen its clarity, depth, and practical relevance.

---

## Round 0.3 · accepted · Accept

Reviewers have confirmed that all their previous comments are addressed in the revised version.

Reviewer 1 ·

Basic reporting

Authors have addressed correctly all the issues pointed out in the previews review.

Experimental design

Authors have addressed correctly all the issues pointed out in the previews review.

Validity of the findings

Authors have addressed correctly all the issues pointed out in the previews review.

Additional comments

Authors have addressed correctly all the issues pointed out in the previews review.

Reviewer 2 ·

Basic reporting

This manuscript is well-revised and can be accepted. The depth of this work is admirable.

Experimental design

This manuscript is well-revised and can be accepted.

Validity of the findings

This manuscript is well-revised and can be accepted.